# *Helicobacter pylori* and *Campylobacter jejuni* bacterial holocytochrome *c* synthase structure-function analysis reveals conservation of heme binding
Tania Yeasmin[1], Susan C. Carroll[2], David J. Hawtof [2,3] & Molly C. Sutherland [2]✉

Heme trafficking is essential for cellular function, yet mechanisms of transport and/or heme interaction are not well defined. The System I and System II bacterial cytochrome *c* biogenesis pathways are developing into model systems for heme trafficking due to their functions in heme transport, heme stereospecific positioning, and mediation of heme attachment to apocytochrome *c*. Here we focus on the System II pathway, CcsBA, that is proposed to be a bi-functional heme transporter and holocytochrome *c* synthase. An extensive structure-function analysis of recombinantly expressed *Helicobacter pylori* and *Campylobacter jejuni* CcsBAs revealed key residues required for heme interaction and holocytochrome *c* synthase activity. Homologous residues were previously identified to be required for heme interaction in *Helicobacter hepaticus* CcsBA. This study provides direct, biochemical evidence that mechanisms of heme interaction are conserved, leading to the proposal that the CcsBA WWD heme-handling domain represents a novel target for therapeutics.

Cytochromes *c* are heme proteins that are encoded by nearly all organisms and function in the context of electron transport chains for cellular respiration and photosynthesis. In bacteria, cytochromes *c* are critical for the ability of organisms to use different energy sources and thus to inhabit multiple environmental niches[1,2]. Cytochromes *c* require the covalent attachment of heme (i.e., cytochrome *c* biogenesis) to a conserved CysXxxXxxCysHis motif via covalent thioether bond formation between the cysteine thiol and heme vinyl groups for proper protein folding and function[3–5]. Cytochrome *c* biogenesis can be accomplished by three pathways: System I (CcmABCDEFGH, *α,γ* Proteobacteria; plant and protozoal mitochondria; Archaea), System II (CcsBA, Gram (+); cyanobacteria; chloroplasts; *ε* Proteobacteria) and System III (HCCS, eukaryotic mitochondria) (reviewed in refs. 6–13). These three pathways use different mechanisms to interact with and position heme for attachment to the CXXCH motif. In bacteria, the cytochrome *c* biogenesis pathways also function to transport heme from the cytoplasmic site of synthesis, across the inner membrane to the site of attachment to apocytochrome *c* in the periplasm[9,11,14].

Here we focus on System II, a bifunctional heme transporter and cytochrome *c* synthase[5,15–17] (Fig. 1) composed of two integral membrane proteins termed CcsBA (also called ResBC)[18–26]. CcsBA has conserved features that are critical for its function. These include four conserved histidines that function as axial ligands to heme[5,15], a tryptophan-rich WWD domain which stereospecifically positions heme for attachment to apocytochrome *c*[15,16] and two large periplasmic domains[16]. The roles for these conserved features were determined by extensive biochemical and structure-function analyses[5,15], as well as cryo-EM structural determination of the *Helicobacter hepaticus* CcsBA[16]. These studies revealed that CcsBA has two heme interaction domains. The transmembrane-heme (TM-heme) domain, consisting of two conserved histidines (TM-His1, TM-His2) and the periplasmic WWD/P-His domain composed of the WWD domain and two conserved histidines located in the periplasm (P-His1, P-His2)[5,15,16]. More recently, a computational study used RoseTTAFold[27] to evaluate the predicted structures of CcsBAs from eight organisms and suggested that the function of the four conserved histidines and the interaction between the WWD domain and heme is highly conserved[17]. However, experimental confirmation of functional conservation predicted by computational studies has not yet been accomplished.

To date, most biochemical characterization of System II has relied on recombinantly expressed *H. hepaticus* CcsBA, the first CcsBA to be recombinantly expressed and affinity purified[5]. Recombinant studies are required as CcsBA is essential in many organisms[18,19,24,25,28–31]. System II is

---

[1]Department of Chemistry and Biochemistry, University of Delaware, Newark, DE, 19716, USA. [2]Department of Biological Sciences, University of Delaware, Newark, DE, 19716, USA. [3]Present address: Department of Biology, University of Virginia, Charlottesville, VA, 22904, USA. ✉e-mail: msuther@udel.edu

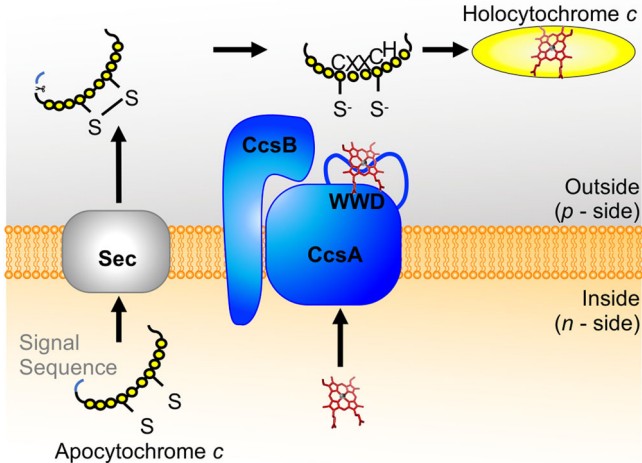

**Fig. 1 | Schematic of System II.** System II, CcsBA (blue), is a bifunctional enzyme, which is proposed to transport heme (red) across the bacterial membrane and attach it to cytochrome *c* (yellow) at a conserved CXXCH motif. CcsBA is encoded as a single ORF that undergoes "natural proteolysis" resulting in CcsB and CcsA components[5] as shown.

most commonly encoded as two proteins (CcsB CcsA), but examples exist where it is encoded as single-fused ORF (CcsBA)[5,9]. System II has presented technical challenges for recombinant expression. The single-fused ORF CcsBA has been amenable to recombinant studies and is stable and functional in *E. coli*[5,32]. In contrast, recombinant expression of System II encoded as two separate proteins (CcsB CcsA) in *E. coli* has not resulted in a functional CcsBA complex (e.g., ref. 33), limiting the ability to perform structure-function analyses. *H. hepaticus* CcsBA has proven to be an excellent model for characterization of CcsBA, however the lack of experimental data for CcsBAs from other organisms has been a major gap in the field.

Here, we describe the recombinant expression, affinity purification and structure-function analysis of CcsBAs from *Helicobacter pylori* and *Campylobacter jejuni*. *H. pylori* is a microaerophilic, Gram-negative bacteria, that is a common cause of chronic gastritis and is considered a causal factor for gastric ulcers and gastric cancer[34]. *C. jejuni* is a microaerophilic Gram-negative bacteria that commonly colonizes bird species and is a leading cause of human bacterial enteritis, which is often acquired through consumption of undercooked chicken[35,36]. The electron transport chains and the ability to switch between types of metabolisms has been implicated in the pathogenesis of both *H. pylori*[37] and *C. jejuni*[38,39]. Therefore, understanding how cytochromes *c*, a critical component of these electron transport chains, are synthesized is of high importance. *H. pylori* only encodes a cytochrome *c* oxidase for growth in the presence of oxygen and a *ccsBA* deletion could not be obtained[28]. While *C. jejuni* encodes a cytochrome *c* oxidase and cytochrome *bd* for growth in the presence of oxygen[28], yet *ccsBA* deletions were still not obtained under microaerobic growth conditions[31]. Thus, demonstrating that *ccsBA* is an essential gene in both these organisms[28,31]. Therefore, recombinant expression is required for genetic studies that investigate the conservation of heme interaction mechanisms and their impact on synthase function. In this study, experimental evidence for conservation of heme interaction and attachment to apocytochrome *c* is demonstrated.

## Results
### Predicted topology and structure of *H. pylori* and *C. jejuni* CcsBA
The well-characterized *H. hepaticus* CcsBA (WP_041309336.1) has 49.5% sequence identity to the *H. pylori* CcsBA (WP_000793250.1) and 40.3% sequence identity to *C. jejuni* CcsBA (CAL35131.1) (Supplementary Fig. 1). Each CcsBA is encoded as a single-fused ORF (Fig. 2a), contains two conserved histidines located in the transmembrane domains (TM-His1, TM-His2), two conserved histidines located in the periplasmic space (P-His1, P-

His2) and a WWD domain (Fig. 2b and Supplementary Fig. 1). For consistency the TM-His1, 2 and P-His1, 2 nomenclature defined for *H. hepaticus* CcsBA was utilized here, and designations were determined by homology to *H. hepaticus* (Supplementary Fig. 1 and Table 1).

*H. pylori* and *C. jejuni* were not included in the recent computational analysis of CcsBA structures[17], thus AlphaFold 3 AI structural predictions[40] of *H. pylori* and *C. jejuni* CcsBA were generated (confidence of predicted structures based on pLDDT are shown in Supplementary Fig. 2a, c) and compared to the *H. hepaticus* cryo-EM structure[16] (Fig. 2c–g). Note that the *H. hepaticus* cryo-EM structure identified two conformations designated as "open" and "closed". The "open" conformation had heme in the TM-heme domain and in the WWD/P-His domain, while "closed" had heme only in the TM-heme domain[16]. The "open" conformation showed more structural similarity to the AlphaFold 3 predictions, thus was used for all comparisons. Based on these structural predications and TMHMM topology analysis[41] *H. pylori* and *C. jejuni* CcsBA each contain 14 transmembrane domains and two periplasmic domains (PD1, PD2), similar to other CcsBAs[17]. Topology schematics for each protein are provided (Supplementary Fig. 2b, d). Recently, a region of the PD1 domain was designated as the "beta cap domain"[17] (Fig. 2c–e, dashed line). While the exact role for the PD1 beta cap domain has not been determined, it was proposed to occlude the CcsBA active site when heme is not present[17]. Here, the beta cap region of PD1 in *H. pylori* and *C. jejuni* was identified by homology to the *H. hepaticus* beta cap region (Fig. 2c–e, dashed line, Supplementary Fig. 2). Superimposition of the *H. hepaticus* cryo-EM structure with the computationally predicated *H. pylori* CcsBA AlphaFold 3 structure indicates a high level of structural conservation (RMSD 1.842 for overlay of complete structures), including positioning of the 14 TM domains and the PD1 and PD2 domains (Fig. 2f). Superimposition of the computationally predicated *C. jejuni* CcsBA AlphaFold 3 structure with the *H. hepaticus* CcsBA cryo-EM structure indicates structural conservation in positioning of the 14 transmembrane domains (RMSD 1.202 for overlay of complete structures), but structural differences in the PD1 domain. *C. jejuni* PD1 is predicted to have an additional beta-sheet region that extends above a structurally well-conserved beta sheet region (Fig. 2g). This beta-sheet extension region was predicted with 70–90% confidence by AlphaFold 3 (Supplementary Fig. 2c). *C. jejuni* CcsBA encodes more than 100 additional amino acids compared to *H. hepaticus* and *H. pylori* in the periplasmic domains (Supplementary Figs. 1 and 2), thus it is unsurprising that structural differences may exist, however direct evidence of these predicted differences awaits structural evidence. Despite the sequence and structural conservation of key features amongst CcsBAs it remains important to demonstrate experimentally that function is conserved.

### Affinity purification of *H. pylori* and *C. jejuni* CcsBA
*H. pylori* CcsBA is 936 amino acids long with a predicted molecular weight of ~106 kDa. *C. jejuni* CcsBA has 1081 amino acids with a predicted molecular weight of ~123 kDa. An N-terminal GST fusion (~26 kDa) was engineered for each CcsBA (Fig. 2a). GST:CcsBAs were recombinantly expressed in *E. coli* C43 Δ*ccm* (lacking the endogenous cytochrome *c* biogenesis genes) and affinity purified. Similar to *H. hepaticus* CcsBA, *H. pylori* (Fig. 3a, b lane 1) and *C. jejuni* (Fig. 3c, d lane 1) CcsBA affinity purifies as two major polypeptides (GST:CcsB' and 'CcsA). Note, *C. jejuni* CcsBA does exhibit additional degradation as indicated in the GST immunoblot (Fig. 3d, lane 1), however it is functional for synthase activity (Fig. 3e). The natural proteolysis of *H. hepaticus* CcsBA occurs at residue 368 and is located in a disordered region near TM5 in the cryo-EM structure[5,16]. Based on this data, the natural proteolysis sites for *H. pylori* and *C. jejuni* were predicted (Fig. 2a indicated by arrows, Supplementary Fig. 2) and the molecular weights for the proteolyzed fragments were calculated (Fig. 2a). As seen in other membrane proteins, specifically in other WWD domain containing proteins, a gel shift occurs resulting in the purified protein running faster than the predicted molecular weight. The proteolysis observed in recombinantly expressed *H. pylori* and *C. jejuni* CcsBAs provides experimental evidence that proteolysis is not limited to *H. hepaticus* CcsBA. Recent RoseTTAFold modeling of CcsBAs from eight

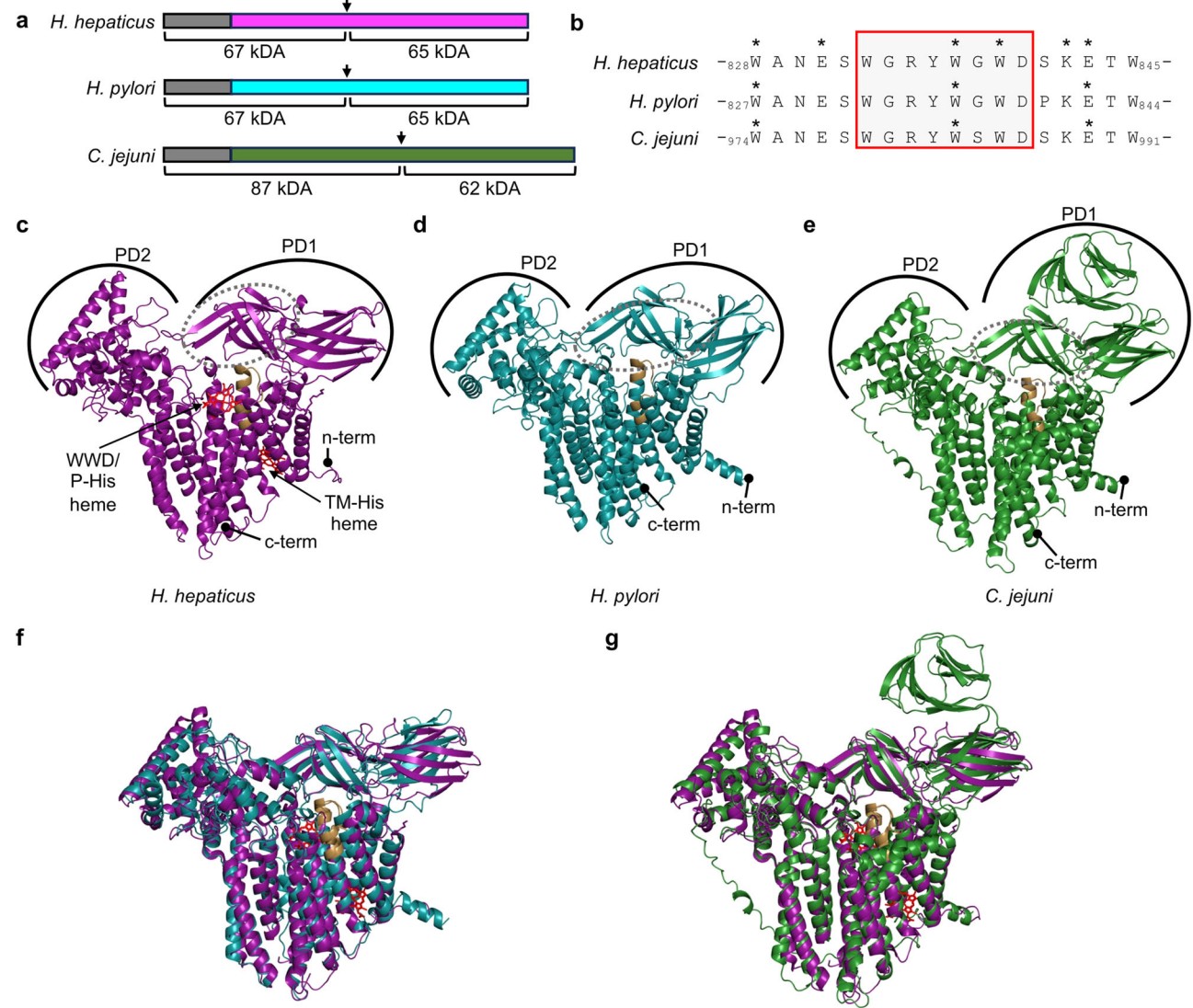

**Fig. 2 | Comparison of bacterial CcsBAs. a** Schematic of CcsBA proteins from *H. hepaticus*, *H. pylori* and *C. jejuni*. N-terminal GST tag is shown in gray. Arrow denotes approximate location of natural proteolysis site. Molecular weights of affinity purified protein polypeptides are shown. **b** Protein sequences of CcsBA WWD domains (*H. hepaticus*: WP_041309336.1, *H. pylori*: WP_000793250.1 *C. jejuni*: CAL35131.1) are shown with conserved region highlighted in red. Asterisks indicate residues shown to form a cysteine/heme crosslink (*H. hepaticus*[15], *H. pylori*, *C. jejuni* this study). **c** Cryo-EM structure of *H. hepaticus* CcsBA (open, PDB 79SY[16]). Heme in red and indicated by arrow with heme interaction domains labeled. **d** AlphaFold 3

predicted structure of *H. pylori* CcsBA (using protein sequence (WP_000793250.1)), **e** AlphaFold 3 predicted structure of *C. jejuni* CcsBA (using protein sequence (CAL35131.1)). PD1 periplasmic domain 1, PD2 periplasmic domain 2, dashed line indicates location of the beta-cap domain, WWD domain in sand, N-term and C-term are indicated. **f** Superimposition of *H. hepaticus* cryo-EM structure (deep purple) and *H. pylori* predicted structure (deep teal), RMSD 1.842.
**g** Superimposition of *H. hepaticus* cryo-EM structure (deep purple) and *C. jejuni* predicted structure (forest green), RMSD 1.202.

other organisms suggests conservation of a disordered region near TM5[17], thus the observed natural proteolysis is unlikely to be an experimental artifact. In fact, a biological role for proteolysis is supported as multiple strategies to eliminate natural proteolysis were unsuccessful. For example, neither expression of *H. hepaticus* CcsBA in a panel of protease deficient *E. coli* strains nor a 20 amino acid deletion encompassing the site of natural proteolysis prevented the observed proteolysis[15].

### Table 1 | Designation of conserved histidines

|               | TM-His1 | TM-His2 | P-His1 | P-His2 |
|---------------|---------|---------|--------|--------|
| *H. hepaticus* | H858    | H83     | H761   | H897   |
| *H. pylori*    | H857    | H86     | H760   | H896   |
| *C. jejuni*    | H1004   | H82     | H904   | H1043  |

### Four conserved histidines are required for heme interaction and synthase function

To evaluate conservation of histidine function in CcsBA, the four conserved histidines of *H. pylori* and *C. jejuni* CcsBA were analyzed. Each of the conserved histidines (see Table 1) were mutated to glycine and an N-terminal GST tag was used for affinity purification. The histidine variants are stable (Fig. 3a–d). Analysis of heme co-purification for each variant was determined via UV-vis spectroscopy using the Soret peak (~412 nm). The *H. pylori* and *C. jejuni* CcsBA TM-His1Gly and TM-His2Gly variants co-purify with ~25% of the heme compared to wildtype (Supplementary Figs. 3a, b and 4a, b). The *H. pylori* P-His1Gly and P-His2Gly variants co-purify with ~50% of wildtype heme levels and *C. jejuni* P-His1Gly and P-His2Gly variants co-purify with approximately wildtype levels of heme (Supplementary Figs. 3a, b and 4a, b). Despite varying levels of heme co-purification, UV-vis spectroscopy analysis determined that the heme

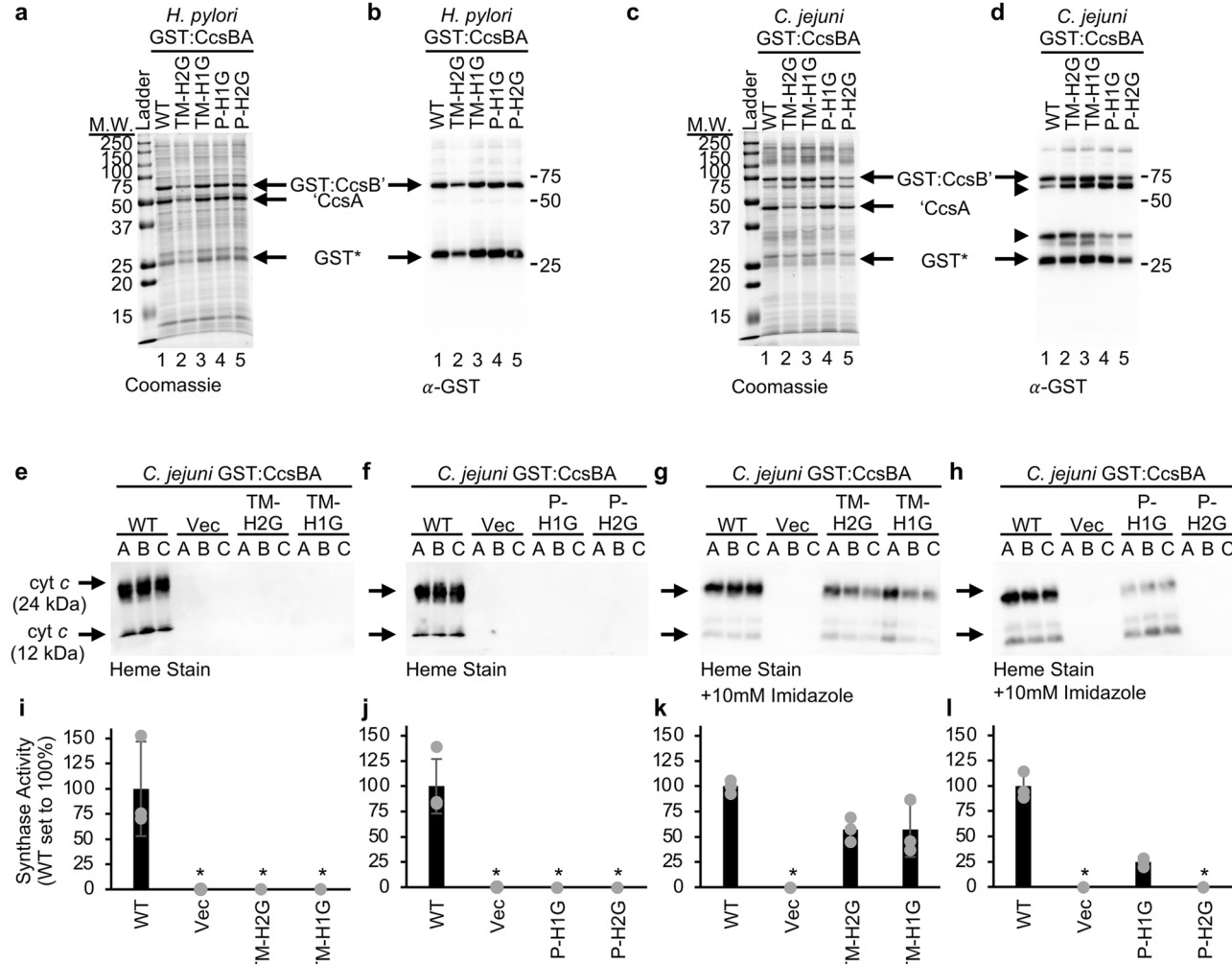

**Fig. 3 | Analysis of conserved histidines.** The four conserved histidines in *H. pylori* and *C. jejuni* CcsBA were individually mutated to glycine. Proteins were affinity purified using an N-terminal GST tag, 5 µg were separated via 12.5% SDS-PAGE and visualized with (**a**, **c**) Coomassie total protein stain and (**b**, **d**) a GST specific antibody. Key molecular weight markers are indicated. CcsBA purifies as two major polypeptides: GST:CcsB' and 'CcsA. '—indicates natural proteolysis. GST*—proteolyzed GST. **e**, **f** The *C. jejuni* CcsBA His→Gly variants were co-expressed with cytochrome $c_4$ *in E. coli* Δccm. Synthase activity (i.e., heme attachment to cytochrome $c_4$) was determined by separation of total cell lysate via SDS-PAGE and heme stain of holocytochrome *c* (24 kDa—full length; 12 kDa—endogenously proteolyzed). **i**, **j** Quantification of heme stained bands was performed with wildtype normalized to 100% synthase function. Three biological replicates, each containing three technical replicates were performed. A representative biological replicate is shown. **g**, **h** Chemical complementation of *C. jejuni* CcsBA His→Gly variants was performed as in (**e**), with supplementation of 10 mM imidazole. **k**, **l** Quantification of heme stained bands was performed as in (**i**) and a representative biological replicate is shown. Error bars show the standard deviation from the mean, dots indicate individual data points. WT wildtype, Vec vector control, * below limit of detection for heme stain.

environment was otherwise similar to wildtype for the His→Gly variants (Supplementary Figs. 3c–g and 4c–g). These results are similar to the analysis of conserved histidines in *H. hepaticus* CcsBA[5,15]. This functional and heme co-purification data supports the model that TM-His1,2 are required for initial heme interaction with CcsBA and in their absence the CcsBA-heme interaction is greatly reduced. In contrast, mutation of P-His1,2 results in wildtype heme levels, suggesting heme is retained in the TM-heme domain as seen the *H. hepaticus* "closed" cryo-EM structure, leading to the model that heme is transferred from the TM-heme domain to the WWD/P-His domain[16].

Previous results determined that the four conserved histidines (Supplementary Fig. 1) are required for CcsBA synthase function[5,15,21,42,43]. To determine if this role in synthase function is conserved for *H. pylori* and *C. jejuni* the TM-His1Gly, TM-His2Gly, P-His1Gly and P-His2Gly variants were recombinantly co-expressed with *Bordetella pertussis* cytochrome $c_4$ in an *E. coli* strain lacking the endogenous cytochrome *c* biogenesis genes (*E. coli* Δccm) and synthase function (i.e., heme

attachment to cytochrome *c*) was monitored via an enhanced chemiluminescence (ECL)-based heme stain[44]. Note, this assay results in full-length holocytochrome $c_4$ (24 kDa) and an endogenously proteolyzed 12 kDa fragment[32]. Under the experimental conditions, two independently constructed wildtype *H. pylori* CcsBA functional strains had an average of ~1/175 synthase function compared to *H. hepaticus* CcsBA (Supplementary Fig. 3h, i) therefore synthase function assays with *H. pylori* were not technically feasible as cytochrome *c* biogenesis levels were near the limit of detection for the heme stain. In *C. jejuni* CcsBA, the four conserved histidines were required for synthase function (Fig. 3e, f, i, j) providing further evidence that the conserved histidines are essential to CcsBA synthase function.

## CcsBA TM-His1, 2 and P-His1 are axial ligands to heme

Histidines often function as axial ligands to heme. It was initially experimentally determined[5] and confirmed by cryo-EM[16] that the *H. hepaticus* CcsBA TM-His act as axial ligands to heme and compose the TM-heme site

(alternatively called the cytoplasmic heme vestibule[16,17]). Recent structural predictions indicate that in the eight CcsBAs examined, TM-His1, 2 are positioned to act as axial ligands to heme[17], as originally shown experimentally in *H. hepaticus*[5]. Here we experimentally determined that *C. jejuni* CcsBA TM-His1, 2 act as axial heme ligands utilizing chemical complementation with exogenous imidazole (Fig. 3g, h). Mutation of histidine to a smaller amino acid, such as alanine or glycine, results in a cavity. When exogenous imidazole is added the imidazole can enter the cavity, restore formation of the heme ligand, and thus protein function. This was originally demonstrated in whale-sperm myoglobin containing a His93Gly mutation and protein crystallization revealed imidazole in the His93Gly cavity bound to the heme iron[45]. Therefore, chemical complementation with imidazole provides evidence that a residue functions in axial liganding of heme. While alanine substitution is often used to biochemically determine the function of a residue, previous studies have demonstrated that alanine substitutions are not robustly chemically complemented with imidazole[5,46], however glycine substitutions can be[15], thus glycine substitutions were used in this study. The *C. jejuni* TM-His1, 2→Gly variants were recombinantly co-expressed with cytochrome $c_4$ in *E. coli* Δ*ccm* with 10 mM imidazole supplementation. *C. jejuni* CcsBA TM-His1Gly and TM-His2Gly are chemically complemented by imidazole (Fig. 3g, k) and therefore function as axial ligands to heme.

Next, the role of P-His1, 2 as axial ligands were assayed as described above. P-His1Gly was chemically complemented, while P-His2Gly was not (Fig. 3h, l). Similar to *H. hepaticus* the *C. jejuni* CcsBA P-His1Gly functions as an axial ligand to heme[15]. The role of P-His2 has not been defined, however lack of chemical complementation with imidazole was also observed in *H. hepaticus* CcsBA[15]. The P-His2 loop was poorly resolved in the *H. hepaticus* CcsBA cryo-EM structure[16] and was poorly predicted in RoseTTAFold analyses[17], suggesting P-His2 is located on a flexible loop. This structural evidence coupled with the lack of imidazole complementation of P-His2Gly[15] (Fig. 3h, l) has led to the proposal that P-His2 undergoes initial ligand exchange with the apocytochrome *c* CXXCH motif histidine[16,17]. Of note, P-His2 in other cytochrome *c* biogenesis WWD domain containing proteins, CcmC[47,48] and CcmF[49], are also located on structurally flexible loops, suggesting this initial ligand exchange by P-His2 may be a common mechanism among these proteins.

## Cysteine/heme crosslinking reveals conservation of heme interaction mechanism in the CcsBA WWD domain

CcsBA is a member of the heme-handling protein (HHP) family which also includes other cytochrome *c* biogenesis proteins CcmC and CcmF from System I[50,51]. This protein family contains a conserved, tryptophan-rich, heme-handling domain called the WWD domain[50] (Supplementary Fig. 5). This domain has been shown to directly bind and stereospecifically position heme in all three HHP proteins[15,16,52,53], thus is considered to be a key feature of the active sites of HHP family members. The CcsBA WWD domains have high sequence conservation (Fig. 2b), but the mechanism of heme interaction has only been experimentally determined in a single CcsBA[15], thus it is essential to experimentally demonstrate if the residues required for heme interaction are conserved.

To identify residues involved with heme interaction in the *H. pylori* and *C. jejuni* CcsBA WWD domains, a cysteine/heme crosslinking approach was taken. Cysteine has a natural propensity to form a covalent bond with heme when the cysteine thiols and heme vinyl groups are in close proximity (Supplementary Fig. 6). Much like heme attachment to the apocytochrome *c* CXXCH motif, other cysteine residues can covalently trap or form a crosslink to heme when engineered into a putative heme interaction domain. The cysteine/heme crosslinking approach has been used to identify heme interacting residues in CcmC[52] and CcmF[53], as well as *H. hepaticus* CcsBA[15]. More recently, cryo-EM structures of CcmABCD[47] and CcsBA[16] confirmed these WWD-heme interactions, demonstrating the utility of cysteine/heme crosslinking to identify heme interaction domains.

The *H. pylori* CcsBA WWD domain is composed of eighteen residues (W827-W844). Single amino acid cysteine substitutions were engineered at each WWD residue in a GST:CcsBA fusion construct. Variants were

recombinantly expressed, affinity purified and 5 µg of purified protein was separated via SDS-PAGE. All WWD cysteine variants were stable (Supplementary Fig. 7a, b). Formation of a cysteine/heme crosslink results in retention of heme at the 'CcsA polypeptide, which encodes the WWD domain, while *b*-type or non-covalently bound heme is separated from the CcsBA polypeptides and runs with the dye front as "free" heme. To determine if a cysteine/heme crosslink was formed, the ratio of 'CcsA:free heme was determined by quantification of a heme stain (Supplementary Fig. 7c). Note that "free" heme can transfer through the nitrocellulose membrane. To ensure accurate quantification of "free" heme, two nitrocellulose membranes are layered for protein transfers and both membranes are heme stained and quantified (see Supplementary Fig. 7f, g). Initially, cysteine variants with a 'CcsA: "free" heme ratio of >2.0 (W827C, A828C, N829C, S831C, K841C, E842C, T843C) or residues that had been shown to form a cysteine/heme crosslink in *H. hepaticus* CcsBA[15] (E830C, W836C, W838C) were further characterized. Subsequent analysis determined that *H. pylori* CcsBA WWD cysteine variants W827C, W836C and E842C form robust cysteine/heme crosslinks (Fig. 4a, b and Supplementary Fig. 7f, g) and are required for heme interaction in the WWD domain. *H. pylori* CcsBA affinity purified proteins have a degradation product that contains the WWD polypeptide as indicated by the heme-stained band (Fig. 4b \*CcsA, Supplementary Fig. 7d–h). Note that covalently bound heme (i.e., from formation of a cysteine/heme crosslink) is retained at the protein polypeptide upon SDS-PAGE, therefore the heme-stained \*CcsA band contains the WWD domain with crosslinked heme.

To determine if the cysteine variants disrupted the overall heme environment of *H. pylori* GST:CcsBA, UV-vis spectral analysis was performed. Cysteine substitution did not alter the overall heme environment (Fig. 4c–f). A caveat to these experiments is that UV-vis spectroscopy provides an average of the heme environment in a sample. It is known that CcsBA retains most of the heme in the cytoplasmic TM-heme site[16], thus this *b*-heme is predicted to account for the majority of the heme signal. Therefore, even if the small amount of crosslinked heme in the WWD domain results in perturbation of the TM-heme environment, UV-vis spectra may not be sensitive enough to detect these changes.

The *C. jejuni* CcsBA WWD domain is composed of residues W974-W991 and similar to the analysis for *H. pylori* described above, 18 single amino acid substitutions were engineered in the GST:CcsBA WWD domain (W974C-W991C). All WWD cysteine variants were stable (Supplementary Fig. 8a, b) and were assessed for formation of the cysteine/heme crosslink (Supplementary Fig. 8c). As described above, residues that had a 'CcsA:free heme ratio of >2.0 (W974C, E989C), or were homologous to *H. hepaticus* CcsBA cysteine/heme crosslinking residues (A975C, W983C, W985C, K988C), or had a visible heme stain band at the 'CcsA polypeptide (N976C, E977C, S978C) were further studied. Subsequent purification and analysis of 'CcsA:free heme ratio identified three cysteine variants that formed a cysteine/heme crosslink (W974C, W983C, E989C) (Fig. 5a, b and Supplementary Fig. 8d). Heme environments of these cysteine variants were similar to wildtype as determined by UV-vis spectroscopy (Fig. 5c–f). The *C. jejuni* cysteine/heme crosslinks at W974C and W983C exhibited a 'CcsA:-free heme ratio of >2.0, but were less robust than the crosslink formed by E989C. Overall *C. jejuni* CcsBA co-purified with lower heme amounts than *H. pylori*, which may account for the lower heme signal at 'CcsA. Together, these analyses have identified conserved residues that are involved with heme binding and positioning in the WWD domain (Fig. 2b \*s). Of note, this study identifies 3 WWD residues that are required for heme interaction across CcsBAs from three organisms, demonstrating that not only does the WWD have high sequence conservation, its mechanism for heme interaction is also highly conserved.

## The CcsBA WWD domain is required for synthase function

The CcsBA WWD domain is required for synthase function[5,15,42,43]. To determine the role of individual *C. jejuni* WWD residues for synthase function, the WWD cysteine variants were co-expressed with cytochrome $c_4$ in *E. coli* Δ*ccm* and synthase function was determined as described above

**Fig. 4 | The *H. pylori* WWD domain directly binds heme.** Single amino acid cysteine substitutions were engineered at each residue in the *H. pylori* WWD domain. Each variant was recombinantly expressed in *E. coli*, affinity purified utilizing the N-terminal GST tag and analyzed for formation of a cysteine/heme crosslink via heme stain. The ratio of CcsA' bound to *b*-type (free) heme was determined and variants with a ratio of >2.0 were selected for further analysis. Three cysteine variants form the cysteine/heme crosslink. Five µg of GST affinity purified protein was separated via SDS-PAGE and assayed for (**a**) protein stability and (**b**) formation of a cysteine heme crosslink. CcsBA purifies as two major polypeptides: GST:CcsB' and 'CcsA. '—indicates natural proteolysis. *CcsA—proteolyzed polypeptide containing the WWD domain with crosslinked heme. GST*—proteolyzed GST. Key molecular weight markers are indicated. **c–f** UV-vis spectral analysis of 75 µg of affinity purified protein. As purified (black) and reduced (red) spectra with key peaks indicated are shown.

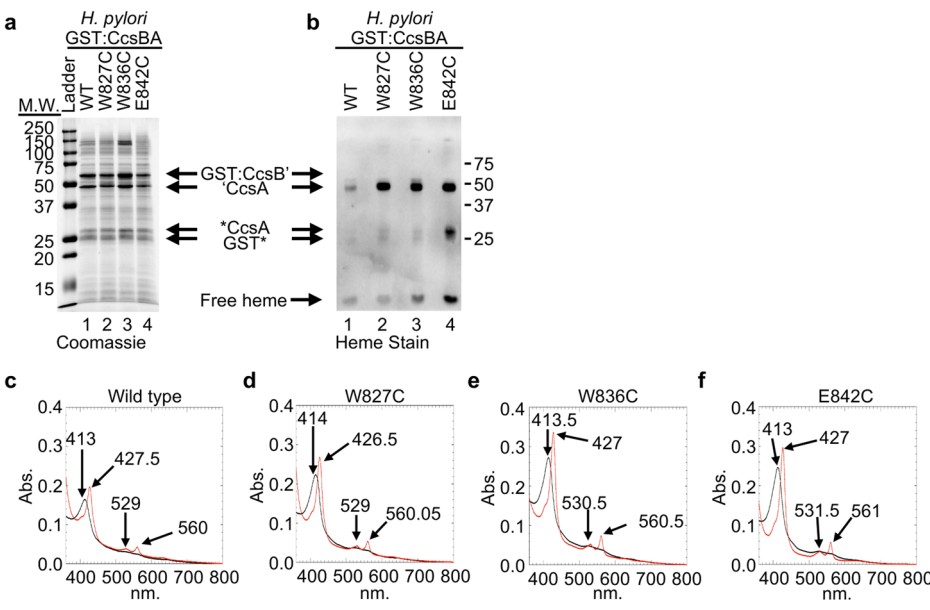

**Fig. 5 | The *C. jejuni* WWD domain directly binds heme and is required for synthase function.** Single amino acid cysteine substitutions were engineered at each residue in the *C. jejuni* GST:CcsBA WWD domain. As described in the Figure, three variants were determined to form the cysteine/heme crosslink. Five µg of GST affinity purified protein was separated via 12.5% SDS-PAGE and assayed for (**a**) protein stability and (**b**) formation of cysteine heme crosslink. CcsBA purifies as two major polypeptides: GST:CcsB' and 'CcsA. '—indicates natural proteolysis. GST*—proteolyzed GST. Key molecular weight markers are indicated. **c–f** UV-vis spectral analysis of 75 µg of affinity purified protein. As purified (black) and reduced (red) spectra with key peaks indicated are shown. **g** *C. jejuni* CcsBA WT or the WWD single amino acid cysteine variants were recombinantly expressed with cytochrome $c_4$ in *E. coli* Δ*ccm* and synthase function was determined as described in Fig. 3e. Three biological replicates, each containing 3 technical replicates were performed. Representative samples are shown. **h** Quantitation of *C. jejuni* CcsBA WT and single amino acid cysteine variant synthase function as described in Fig. 3f. WT normalized to 100% synthase function. Error bars show the standard deviation from the mean, dots indicate individual data points. WT wildtype, Vec vector control, *below limit of detection for heme stain.

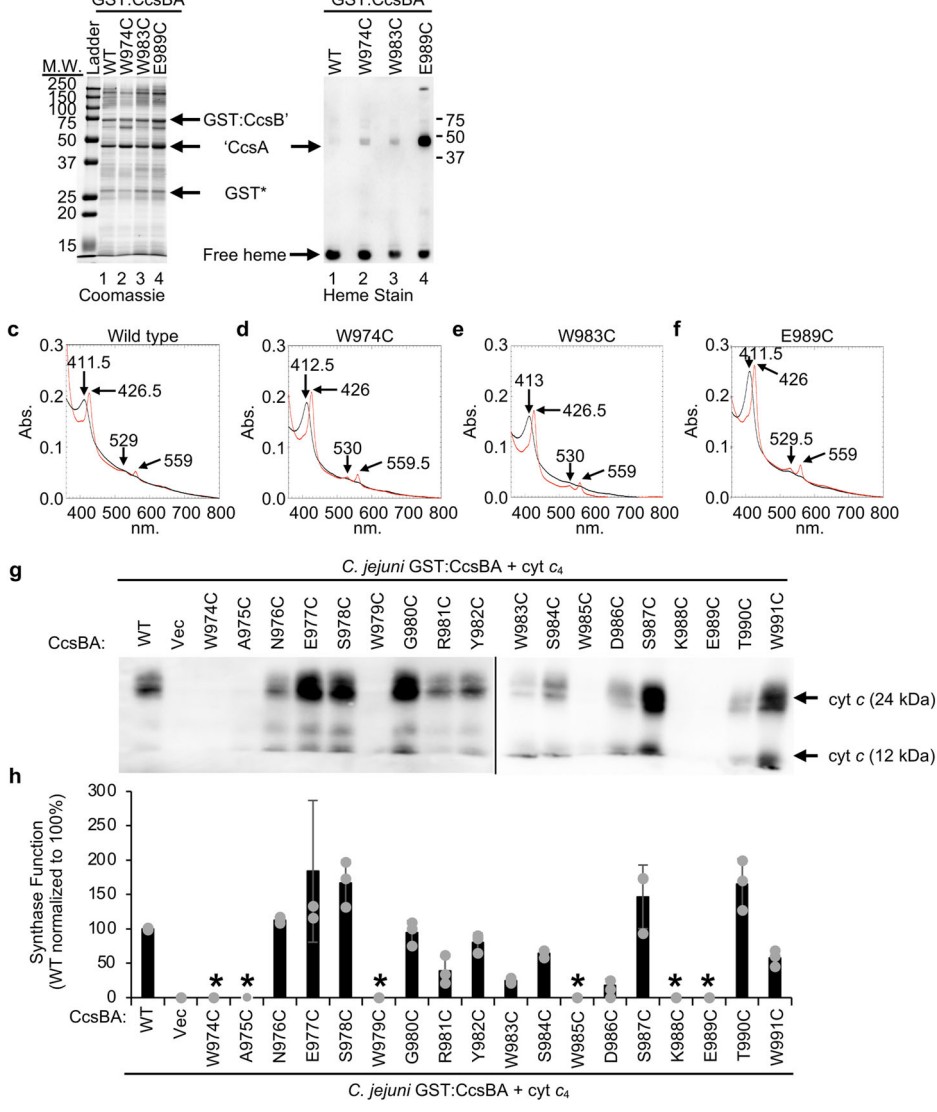

(Fig. 5g, h and Supplementary Fig. 9). Six residues (W974C, A975C, W979C, W985C, K988C, E989C) were essential for synthase function, defined as less than 5% synthase function compared to WT. Two residues (W983C, D986C) had severely reduced function, defined as 5-25% synthase function compared to wildtype. Three residues (R981C, S984C, W991C) had partial function, defined as 25-80% of wildtype synthase function. Seven residues (N976C, E977C, S978C, G980C, Y982C, S987C, T990C) had wildtype synthase function, defined as >80% of wildtype synthase function, thus are not required for *C. jejuni* CcsBA synthase function. While there are differences in the requirement of specific residues for synthase function across CcsBAs, this analysis identified three conserved residues that are required for synthase function across *H. hepaticus* and *C. jejuni* CcsBAs (*C. jejuni*: W974C, A975C, W979C; *H. hepaticus*: W828C, A829C, W833C[15]). Of note, a limited analysis of the *Chlamydomonas reinhardtii* CcsBA WWD domain was undertaken and residues homologous to *C. jejuni* W974 and W979 were also required for synthase function[42]. These results demonstrate that not only is the WWD domain required for synthase function and thus constitutes the active site of CcsBA, but we have also determined residues that contribute specifically to synthase function and heme binding.

## Discussion

Here a comprehensive structure-function analysis of heme binding in *H. pylori* and *C. jejuni* CcsBAs was undertaken to determine if heme binding mechanisms are conserved in System II bacterial cytochrome *c* biogenesis. Previous work defined a TM-heme site on the cytoplasmic face of *H. hepaticus* CcsBA, composed of TM-His1 and TM-His2[5,15,16]. We have experimentally shown that TM-His1,2 are required for heme binding in two additional CcsBAs, as His→Gly mutations drastically reduced heme co-purification (Supplementary Figs. 3a, b and 4a, b). Further, TM-His1,2 in *C. jejuni* CcsBA act as axial heme ligands (Fig. 3g, k). Experimental evidence that the *H. pylori* CcsBA TM-His1,2 are required for synthase function and act as axial heme ligands was not obtained due to low levels of synthase function (Supplementary Fig. 3h, i). However, based on conservation of TM-His1,2 and the experimental evidence from *H. hepaticus* CcsBA[5,15], *W. succinogenes* CcsBAs[43], *Chlamydomonas reinhardtii*[21,42] and *C. jejuni* CcsBAs (Fig. 3e, g) we predict *H. pylori* TM-His1,2 will function as axial ligands to heme. Conservation of this TM-heme binding domain was predicted by computational analyses of eight CcsBAs[17]. AlphaFold 3 can predict protein-molecule interactions[40] and was used to predict heme interaction with *H. pylori* and *C. jejuni* CcsBA. These structural predictions were compared with the *H. hepaticus* CcsBA cryo-EM and show positioning of TM-His1,2 is similar, supporting the chemical complementation data that these residues act as axial heme ligands (Fig. 6a–c).

Heme is proposed to be trafficked from the TM-heme site to the external WWD/P-His domain via a heme channel composed of the 4 TM domains (TM10, TM11, TM12, TM13) surrounding the WWD domain, termed the WWD domain "core region". The structural architecture of the "core region" is conserved in *H. hepaticus*, *H. pylori* and *C. jejuni* CcsBAs (Fig. 6d), as well as other computationally predicted CcsBA structures[17]. Here we have experimentally shown that P-His1 and P-His2 of *C. jejuni* CcsBA are required for synthase function (Fig. 3f, j) and P-His1 is an axial ligand to heme (Fig. 3h, l). The WWD domain has now been shown to bind and stereospecifically position heme in three CcsBAs[15] (Figs. 4b and 5b), as well as the other two members of the HHP family via cysteine/heme crosslinking[52,53]. This study demonstrates that the mechanism of heme binding within the WWD domain is highly conserved in CcsBAs and three residues of high importance were identified (Figs. 2b *s and 6e–g). Formation of the cysteine/heme crosslink confirms that these residues are in close proximity to the heme vinyl groups (Supplementary Fig. 10a, d–g). AlphaFold 3[40] predicted heme interaction with *H. pylori* and *C. jejuni* CcsBA WWD domains (Supplementary Fig. 10b, c) were compared to heme positioning in *H. hepaticus* CcsBA determined by cryo-EM (Supplementary Fig. 10). *C. jejuni* predicted heme positioning is similar to *H. hepaticus* with both heme vinyl groups located near a cysteine/heme crosslinking residue. *H. pylori* predicted heme positioning did not correspond to the *H. hepaticus*

CcsBA heme positioning (Supplementary Fig. 10b), thus confirmation of heme vinyl group positioning in the WWD domains of *H. pylori* and *C. jejuni* CcsBA awaits structural determination. The cysteine/heme cross-linking results can also provide insight into positioning of the apocytochrome *c* CXXCH motif, which also must be close to heme vinyl groups for thioether bond formation. Computational studies predicted two binding pockets for the CXXCH motif in *H. hepaticus* CcsBA, called C1 (composed of the C-terminal WWD domain, specifically *H. hepaticus* residues D840, S841, K842, E843) and C2 (composed of N-terminal WWD domain, specifically *H. hepaticus* residues W828, A829, S832, W833)[17]. Of note, two of the three conserved cysteine/heme crosslinking residues determined here (homologous to *H. hepaticus* W828, E843), reside within those predicted CXXCH binding pockets. Utilizing the *H. hepaticus* cryo-EM structure with heme in the WWD domain W828C and E843C also have the shortest distance to the heme vinyl groups (Supplementary Fig. 10), thus it is unsurprising that cysteine substitutions at these residues form a cysteine/heme crosslink. Together this modeling and experimental evidence support that heme is positioned within the WWD domain to subsequently form thioether bonds to the CXXCH motif of apocytochrome *c*. In conclusion, this study demonstrates that heme interaction in the TM-heme, as well as positioning in the WWD/P-His domain of the bacterial cytochrome *c* synthase, CcsBA, is highly conserved.

The conservation of heme binding residues in the cytoplasmic TM-heme site and the periplasmic WWD/P-His heme domain determined here are important to our basic biological understanding of cytochrome *c* synthase function. Conservation of heme binding in the active site of CcsBA from multiple bacteria leads to the proposal that CcsBA could be a target for novel inhibitors. Early studies showed that addition of exogenous alternative metal porphyrins (e.g., ZnPPIX) inhibited cytochrome *c* biogenesis by recombinantly expressed System II and System I pathways[54], demonstrating that inhibition of these pathways is feasible. CcsBA is an attractive target as it is essential in pathogens including *H. pylori* and *C. jejuni*[18,19,24,25,28–31], thus targeting the synthase could inhibit pathogen growth. Additionally, differences in substrate recognition exist between CcsBA and the human cytochrome *c* synthase, HCCS, therefore cross-reactivity is unlikely. For example, in vivo studies suggested and recent in vitro reconstitution confirmed that HCCS requires the apocytochrome *c* CXXCH motif plus the preceding N-terminal alpha helix for substrate recognition[55–58]. In contrast, CcsBA has been shown to mature cytochromes *c* from various bacterial species, as well as the human cytochrome *c*[59,60] and in vitro reconstitution demonstrated CcsBA heme attachment to a minimal 9-mer peptide (KCSQCHTVE)[55]. While there is a general need for novel therapeutics, *H. pylori* and *Campylobacter* species are designated as high-priority pathogens for new therapeutics due to their clarithromycin and fluroquinolone resistance, respectively[61,62], thus novel antibiotic targets are of high importance. This study demonstrates conservation of WWD residues in heme binding (Figs. 4b and 5b) and synthase activity[5,15,42,43] (Fig. 5g, h) across CcsBAs. Therefore, we propose CcsBA, specifically the WWD domain is potentially an attractive target for novel therapeutics.

## Methods

### Bacterial growth conditions

*Escherichia coli* strains were grown in Luria-Bertani broth (LB; Difco) at 24 °C and 240 rpm for protein purifications or 37 °C and 200 rpm for functional assays with the following selective antibiotics and inducing reagents at the indicated concentrations: carbenicillin, 50 μg/ml; chloramphenicol, 20.4 μg/ml; isopropyl -D-1- thiogalactopyranoside (IPTG; Gold Biotechnology), 1.0 mM or 0.1 mM; L-arabinose (Gold Biotechnology), 0.2% (wt/vol).

### Construction of strains and plasmids

All cloning was performed using *E. coli* NEB-5α. Construction of plasmids and generation of single amino acid substitutions is described in the Supplementary Methods. A complete list of strains, plasmids, and primers is provided in Supplementary Table 1.

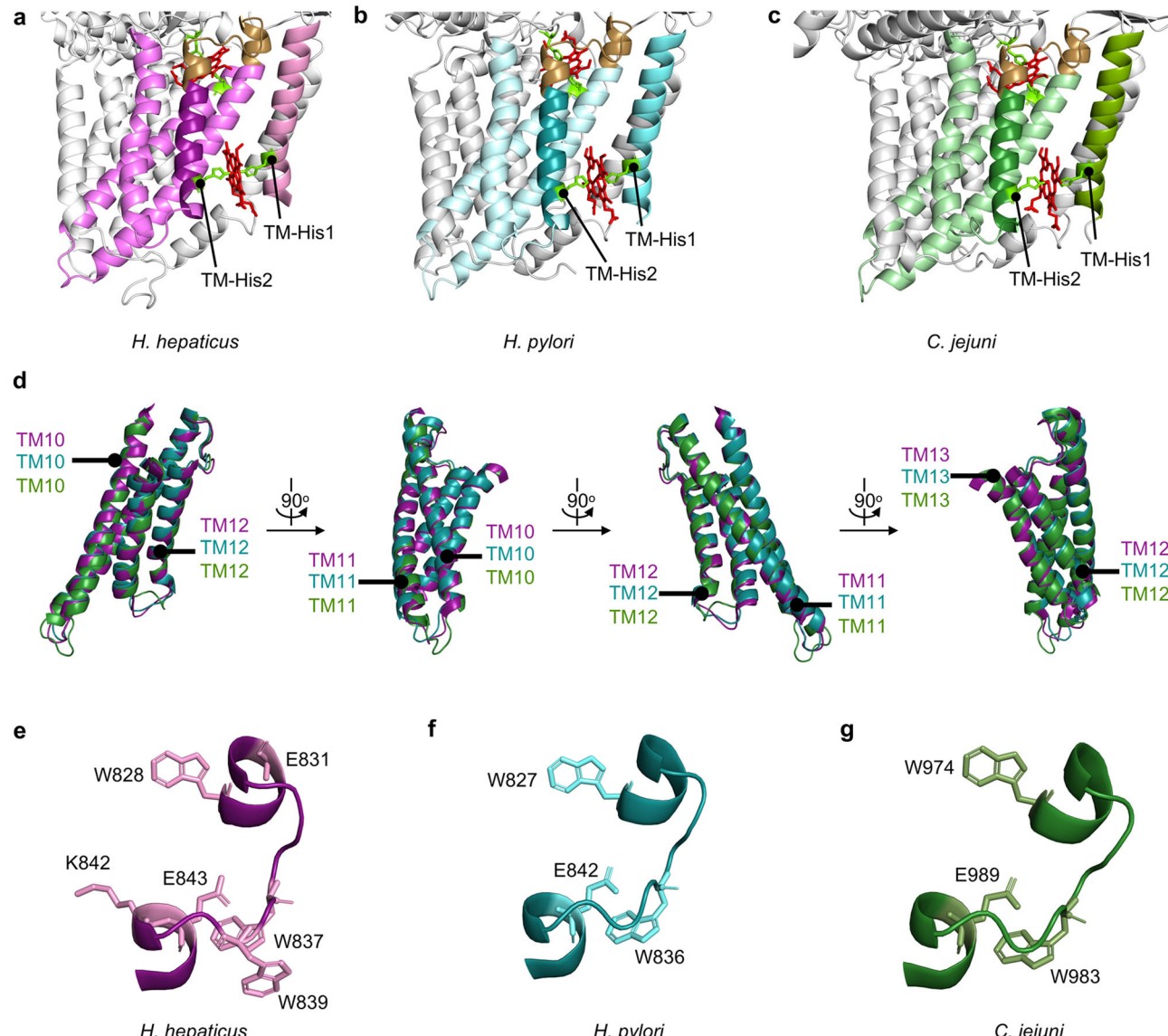

**Fig. 6 | Heme binding mechanism is conserved in CcsBA.** The cryo-EM structure of *H. hepaticus* CcsBA (deep purple, open, PDB 79SY[16]), AlphaFold 3 predicted structure of *H. pylori* CcsBA (deep teal) and AlphaFold 3 predicted structure of *C. jejuni* CcsBA (forest green) were used to examine heme binding in CcsBA. **a–c** TM-His1 and TM-His2 (lime green) are positioned to act as axial ligands to heme localized in the TM-Heme site. Note, TM1 and TM2 are hidden for visualization of TM-His positioning. **d** The "WWD core", consisting of the WWD domain and four surrounding transmembrane domains of each CcsBA were superimposed and 90° rotation is shown. TMs are labeled. **e–g** The WWD domain is shown and residues that form a cysteine/heme crosslink are labeled and shown with amino acid structure.

## CcsBA structural modeling

The *H. pylori* (strain ATCC 700392/26695) CcsBA and *C. jejuni* (strain ATCC 700819/NCTC 11168) CcsBA predicted structures were obtained using the AlphaFold 3[40] launched on 2024-05 to predict protein structures and protein-ligand (heme) interactions. For both the organisms, most of the predicted structures indicated a high level of model confidence (per residue confidence score, pLDDT, see Supplementary Fig. 2a, c). The PDB files of *H. pylori* & *C. jejuni* (obtained from AlphaFold 3 prediction) and *H. hepaticus* CcsBA open (PDB: 7S9Y[16]) conformation were uploaded to PyMOL (version 2.5.2). Then these structures were superimposed to give the root mean standard deviation (RMSD) values between them.

## Protein purifications

*E. coli* strain C43 was used for protein expression and GST affinity purifications of GST:CcsBA were performed as previously described[15]. Details are provided in Supplementary Methods.

## Heme staining, immunoblotting and quantification

Heme staining was performed as previously described utilizing an ultra-sensitive, long duration ECL-based development (Thermo Scientific FEMTO kit)[15,44,53]. Immunoblots were performed using 5 µg of affinity purified proteins and α-GST (1:75000) (Invitrogen, PA1-982A), primary, protein A peroxidase (Millipore Sigma, P8651) as a secondary label and a standard ECL development. Imaging was performed on Sapphire Biomolecular Imager (Azure Biosystems) and quantified with imaging software (Azure Spot v.2.2.167).

## UV-visible absorption spectroscopy

UV-visible absorption spectra were collected with a UV-1900i spec and LabSolutions software (Shimadzu; LabSolutions UV-vis version 1.10) as described in ref. 15. Briefly, 75 µg of affinity purified protein was used to collect spectra from 360–800 nm. Quantitation of total heme levels used the Soret region (~411–414 nm). Sodium hydrosulfite powder (Sigma 157-953)

was used to reduce protein spectra. Note that initial CcsBA studies did not accomplish complete reduction of CcsBA[5], subsequent studies determined that in order to obtain a completely reduced CcsBA spectra, excess sodium hydrosulfite is required[15].

### Functional (heme attachment) assays

The indicated CcsBA variants were co-expressed with cytochrome $c_4$:His (pRGK332) in strain C43 $\Delta ccm$::Kan$^R$. Heme staining was performed on all samples to confirm the heme attachment function of CcsBA variants as described previously[15,32,63]. Details are provided in the Supplementary Methods.

### Statistics and reproducibility

Statistical analysis of data presented in graphs was performed in Microsoft Excel. All protein purifications were performed independently and a minimum of three times. All UV-vis spectral analysis was performed with independent protein preparations a minimum of two times. All functional experiments were performed independently and in triplicate with biological and/or technical replicates, as described in figure legends.

### Reporting summary

Further information on research design is available in the Nature Portfolio Reporting Summary linked to this article.

## Data availability

Uncropped gel images are provided in Supplementary Figs. 11 and 12. Source Data for Figs. 3e–i and 5g, h are provided in Supplementary Data 1 and 2.

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

## Acknowledgements

We thank Robert G. Kranz for generous use of strains and plasmids (indicated by pRGK numbers in Supplementary Table 1). We acknowledge Robert G. Kranz and the Washington University in St. Louis Fall 2010 Bio437 class for the cloning of *C. jejuni* GST:CcsBA. We thank Brett A. Graver for cloning assistance; Ameeta Balaji and Donna R. Price for technical assistance. Research reported in this publication was supported by University of Delaware start-up funds, the National Institute of General Medical Sciences of the National Institutes of Health under Award Number R35GM142496 to M.C.S. and P20GM104316. The content is solely the responsibility of the authors and does not necessarily represent the official views of the National Institutes of Health.

## Author contributions

Tania Yeasmin: investigation, formal analysis, validation, writing—original draft preparation, writing—review and editing; Susan C. Carroll: investigation, formal analysis, validation, writing—original draft preparation, writing—review and editing; David J. Hawtof: investigation, formal analysis, validation, writing—review and editing; and Molly C. Sutherland: conceptualization, formal analysis, funding acquisition, methodology, resources, supervision, visualization, writing—original draft, writing—review and editing.

## Competing interests

The authors declare no competing interests.
