## [Peer review file · Communications Biology]

Reviewers' comments:

Reviewer #1 (Remarks to the Author):

The manuscript "Helicobacter pylori and Campylobacter jejuni bacterial holocytochrome c synthase structure-function relation analysis reveals conservation of heme binding" provides a very sound and thorough biochemical characterization of the CcsBA membrane complex (system II pathway), which acts as a cytochrome c synthase and is of high importance, due to the fact that heme has to be inserted correctly in the cytochromes. The scientific gain of such studies on the purified enzymes is very high, as only then true mechanistic questions can be addressed.

The authors convincingly show that mechanisms of heme interaction are conserved, putting the spotlight on the WWD heme handling domain.

There are only a few minor things the authors should consider in a revised version:

1) lines 297 and following: It is surprising that the overall heme environment is not impacted concerning UV-vis spectral features upon cysteine substitution. This is explained by the fact that with this method always the average of all spectra is observed. This is true, but nevertheless also in UV-vis spectroscopy there are much more possibilities to obtain meaningful data that can be interpreted reliably.

2) In general, spectra in Figures 4, 5, S3, S4 could be presented in a more comprehensive way. The visible region should be expanded, and interpreted. Most as far as I can judge, most investigated samples are in a six-coordinated low spin (most likely bis-histidine, or bis-imidazole) state, which does fit to the storyline. Nevertheless, this coordination should actually change to a high spin species in the His-Gly variants prior to imidazole addition, this is not the case in the presented spectra. Possibly other effects of structural rearrangement come into play. I understand that this detailed spectral analysis might be out of scope of this work and it probably should be addressed in a follow up study, also in combination with other high resolution spectroscopic methods (electron paramagnetic resonance or resonance Raman spectroscopy). Also I may have missed the exact conditions of how the reduced

samples were obtained. The sodium dithionite peak at 315 nm is not visible but appears to be very large, indicating a big excess, which also has a significant acidification effect.

3) line 227: typo in imidazole

Reviewer #2 (Remarks to the Author):

The maturation of cytochrome c is tightly regulated to ensure proper heme attachment and function. Bacterial cytochrome c maturation involves two primary systems: System I (CcmABCDEFGH) and System II (CcsBA), which are utilized by different bacterial species. In 2021, the Kranz group reported the cryo-EM structure of CcsBA from *Helicobacter hepaticus*, revealing two heme binding sites (TM-heme site and P-heme site) in the open conformation. Additionally, in 2018, Sutherland, Kranz, and colleagues studied the heme binding WWD motif using a cysteine/heme crosslinking tool to trap endogenous heme. Here, the Sutherland group took a further step in investigating the three heme binding sites in CcsBA of *Helicobacter pylori* and *Campylobacter jejuni* using similar approaches. The manuscript is nice to read, it is well written and clear. Below are some comments that should be addressed prior publication.

Major comments:

- 1) It is interesting that *C. jejuni* contains an additional beta-sheet domain. According to the AlphaFold predicted structure, it is possible to construct a version with the additional domain deleted and experimentally evaluate its effect on cytochrome c maturation. This would be a valuable addition alongside sequence alignment and structure prediction.
- 2) Since the AF3 server is available, adding predictions of CcsBA with heme binding and comparing them with *H. hepaticus* structure would be a valuable addition to Figure 6.
- 3) While it is enticing to suggest that the WWD domain could serve as a novel therapeutic target, the lack of supporting data necessitates caution. The author should consider either removing this section from the conclusion or providing preliminary data. For example, demonstrating how a compound might bind to the WWD motif and inhibit cytochrome c maturation could strengthen the statement.

Minor points:

- 1) L99, structural predictions are not “determined”, should only be used for experimental determination.
- 2) L189, missing percentage number?
- 3) L242, The Ilcu et al 2023 CcmABCD/E paper should be cited.
- 4) L285, Figure 4b and supplementary Figure 7d are identical. Same problem in Figure 5b and supplementary Figure 8d
- 5) Supplementary Figure 9, quantify the data and present it also in histogram similar to that in Figure 3.

Reviewer #3 (Remarks to the Author):

The manuscript entitled “*Helicobacter pylori* and *Campylobacter jejuni* bacterial holo-cytochrome c synthase...” by Yeasmin, T. et al., is on the heme binding sites of the CcsBA protein(s) from these species with System II cytochrome c biogenesis. The authors provide experimental support for the functions of structurally conserved features of these proteins using biochemical approaches. The experimental work was carried out very well, and the conclusions are well supported. However, similar studies (Ref 15) and the cryoEM structure (ref 16) of the CcsBA complex from *Helicobacter hepaticus*, a homologous species, have been reported earlier. Although this study could be seen as a repetition of what has been done earlier with *H. hepaticus*, it might be argued that providing experimental support for functional conservation of predicted structures is important in the case of *H. pylori* and *C. jejuni* as this information may eventually provide more reliable therapeutic target(s), making this study worthy of publication.

The following mostly minor modifications may further improve the manuscript.

1- Fig. 1 legend: for clarity, mention that CcsBA is a single protein that undergoes “natural proteolysis” yielding the CcsB and CcsA components as shown and add a reference.

2- p. 4, l.57: To help the reader, perhaps add a sentence describing the basis of the computational approach used to evaluate the structural conservation between the predicted structures (e.g., superimposition of a cryoEM derived structure with those computationally (Rosetta, AlphaFold...) predicted structures, and determination of the RMSD values around the conserved features

3- p. 4, l.69: clarify "...CcsB CcsA have not been functional..."; I guess the authors mean that: the recombinant expression of the CcsB and CcsA as two separate proteins in *E. coli* has not yielded a functional CcsBA complex (if known, due to some reason like not assembled, not stoichiometric, not stable, not soluble, etc etc..), unlike CcsBA as a single fused ORF.

4- p. 5, l.71: "... lack of analysis ... a major gap": perhaps add "...lack of experimental data..."

5- Figure 2 and other figures: a- please make consistent the figure legends and related text using capital letters with the figure panels using small cases. b- if possible, provide an overall (or regional) RMSD value for deviations between the compared structures to give the reader a feeling about how good is the conservation between the structures. For easier visualization, is it possible to color the heme moieties on *H. hepaticus* in black, or point them out by arrows? Also, indicate the α -cap region of PD1 with thicker dotted line to ease noting its location on the structures?

6- Just a curiosity: can the extra ~100 AAs long beta-sheet portion of *C. jejuni* PD1 domain be deleted (at least in *E. coli*) and if so, is it functional? Is it plausible that CcsBA is interacting with another protein in *C. jejuni*?

7- p. 9, l.162: to support this suggestion, has any mutagenesis of the predicted proteolytic cleavage site been performed in any species, and what was the outcome?

8- Fig. S3, legend, Table S1: clarify the genotype(s) of MS1175 and MS1025 (wt or mutant)

9- p. 10, l.205 on: a- Considering that *H. pylori* (I assume wt) CcsBA activity cannot be determined using *B. pertussis* cytochrome c4, then what is the proof that the function of the 4 His conserved in this species? Is it by inference to *C. jejuni* and *H. hepaticus*? If so, clearly state this point. b- Could a cytochrome distinct from c4 be more suitable as a substrate for *H. pylori*?

10- p. 11, l.215: a- add a reference to “cytoplasmic heme vestibule”? b- l. 227, “imidazole” spelling?

11- Fig. 4 and Fig. S7: What is the heme-stained band visible with E842C mutant running between CcsA and GST*, and what is GST*?

12- p. 14, l.299-303: a- Clarify and modify the sentence: if the amount of crosslinked-heme in the WWD domain is very small, even this perturbs the TM-heme environment, UV-Vis spectra may not be sensitive enough to show it. b- is it possible to detect by pyridine/hemochrome the XL-heme derivatives, which should act as a cyt c? As this method would get rid of cyt b, could it provide a better means of quantification for CcsA~XLheme than that used here (ratio)?

13- p. 17, l.346, please recheck carefully the number of the residues mentioned in the text and in Fig. 5 (W893 or W983? S978 or S988?).

14- Introduction: Could you better explain why Ccs system is essential for *H. pylori* and *C. jejuni*, and this under which specific growth conditions? Is this due to the lack of the cytochrome c containing respirator pathway with a single cbb3-type cytochrome oxidase and absence of any other only b-type cytochrome containing alternative respiratory branch? Is CcsBA also essential in *H. hepaticus*, and if not explain why? Perhaps emphasizing this point might add more value to this work and save it from appearing as a repetition of what has been done earlier?

15- Fig. 5c: Explain why two bands correspond to cyt c4 of *B. pertussis* (12 and 24 kDa, dimerization or proteolysis?) seen in this and other figures.

No obvious issue with statistics and reproducibility of the work.

Response to Reviewers for COMMSBIO-24-2120 (*Helicobacter pylori* and *Campylobacter jejuni* bacterial holo-cytochrome c synthase structure-function analysis reveals conservation of heme binding). Thank you to the reviewers for your careful reading and thoughtful comments on our manuscript, which have helped to improve and clarify the manuscript. Point-by-point responses are provided, changes to manuscript are highlighted in yellow and page/line numbers of changes are provided.

Reviewer #1 (Remarks to the Author):

The manuscript "Helicobacter pylori and Campylobacter jejuni bacterial holo-cytochrome c synthase structure-function relation analysis reveals conservation of heme binding" provides a very sound and thorough biochemical characterization of the CcsBA membrane complex (system II pathway), which acts as a cytochrome c synthase and is of high importance, due to the fact that heme has to be inserted correctly in the cytochromes. The scientific gain of such studies on the purified enzymes is very high, as only then true mechanistic questions can be addressed.

The authors convincingly show that mechanisms of heme interaction are conserved, putting the spotlight on the WWD heme handling domain.

There are only a few minor things the authors should consider in a revised version:

1) lines 297 and following: It is surprising that the overall heme environment is not impacted concerning UV-vis spectral features upon cysteine substitution. This is explained by the fact that with this method always the average of all spectra is observed. This is true, but nevertheless also in UV-vis spectroscopy there are much more possibilities to obtain meaningful data that can be interpreted reliably.

Response: Thank you for this comment. We previously have seen that cysteine substitutions do not impact the overall heme environment to the extent that is observable via UV-vis spectra in multiple cytochrome c biogenesis proteins (CcsBA - Sutherland et al. 2018. mBio: doi.org/10.1128/mbio.02134-18; CcmC – Sutherland et al 2018. J Mol Biol: doi: 10.1016/j.jmb.2018.01.022; CcmF – Grunow et al. 2023. mBio: doi: 10.1128/mbio.01509-23). We have proposed that this is likely due to a low amount of cysteine/heme crosslink formation, relative to the total heme present in affinity purified protein preparations. To address this comment and reviewer 3, pt 12 the following statement was added to clarify this point, please see pg 16, line 328:

"...Therefore, even if the small amount of crosslinked heme in the WWD domain results in perturbation of the TM-heme environment, UV-vis spectra may not be sensitive enough to detect these changes."

2) In general, spectra in Figures 4, 5, S3, S4 could be presented in a more comprehensive way. The visible region should be expanded, and interpreted. Most as far as I can judge, most investigated samples are in a six-coordinated low spin (most likely bis-histidine, or bis-

imidazole) state, which does fit to the storyline. Nevertheless, this coordination should actually change to a high spin species in the His-Gly variants prior to imidazole addition, this is not the case in the presented spectra. Possibly other effects of structural rearrangement come into play. I understand that this detailed spectral analysis might be out of scope of this work and it probably should be addressed in a follow up study, also in combination with other high resolution spectroscopic methods (electron paramagnetic resonance or resonance Raman spectroscopy). Also I may have missed the exact conditions of how the reduced samples were obtained. The sodium dithionite peak at 315 nm is not visible but appears to be very large, indicating a big excess, which also has a significant acidification effect.

Response: We thank the reviewer for careful analysis of the UV-vis spectra data. We have answered each point below:

- *Expansion of the visible region: We obtain spectra using scanning UV-vis from 360-800 nm, therefore expansion of the spectra was not possible. We will keep this in mind for the future.*
- *His → Gly variants: It has been previously proposed that CcsBA likely retains most heme in the TM heme domain, liganded by TM-His 1,2 (Sutherland et al. 2018. mBio: doi.org/10.1128/mbio.02134-18, Sutherland et al. 2021. eLife: https://doi.org/10.7554/eLife.64891; Mendez et al. 2022. Nat Chem Biol: doi: 10.1038/s41589-021-00935-y), thus for the P-His → Gly variant spectra the majority of the heme spectra is still obtained from the TM-His1,2 liganded heme. For the TM-His → Gly variants there is a drastic reduction in heme co-purification, thus a reliable in-depth analysis may not be feasible.*
- *Reduced UV-vis spectra: Reduced spectra were obtained by addition of sodium hydrosulfite/sodium dithionite. The reviewer is correct that the spectra indicates an excess of reductant, this was previously shown to be required for full reduction of CcsBA proteins. We have added the following statement to the material and methods to describe this, please see pg 24, line 519:*

“...Note that initial CcsBA studies did not accomplish complete reduction of CcsBA⁵, subsequent studies determined that in order to obtain a completely reduced CcsBA spectra, excess sodium hydrosulfite is required¹⁵.”
- *-Advanced/detailed spectral analysis: Thank you for this suggestion, we agree this would be informative, but outside the scope of the current study.*

3) line 227: typo in imidazole

Response: Thank you for catching this, it has been corrected. Please see pg 12, line 246: “...robustly chemically complemented with imidazole^{5,45}.”

Reviewer #2 (Remarks to the Author):

The maturation of cytochrome c is tightly regulated to ensure proper heme attachment and function. Bacterial cytochrome c maturation involves two primary systems: System I

(CcmABCDEFGH) and System II (CcsBA), which are utilized by different bacterial species. In 2021, the Kranz group reported the cryo-EM structure of CcsBA from *Helicobacter hepaticus*, revealing two heme binding sites (TM-heme site and P-heme site) in the open conformation. Additionally, in 2018, Sutherland, Kranz, and colleagues studied the heme binding WWD motif using a cysteine/heme crosslinking tool to trap endogenous heme. Here, the Sutherland group took a further step in investigating the three heme binding sites in CcsBA of *Helicobacter pylori* and *Campylobacter jejuni* using similar approaches. The manuscript is nice to read, it is well written and clear. Below are some comments that should be addressed prior publication.

Major comments:

1) It is interesting that *C. jejuni* contains an additional beta-sheet domain. According to the AlphaFold predicted structure, it is possible to construct a version with the additional domain deleted and experimentally evaluate its effect on cytochrome c maturation. This would be a valuable addition alongside sequence alignment and structure prediction.

Response: We thank the reviewer and agree that analysis of the periplasmic domains, in particular the extended beta-sheet domain predicted in the C. jejuni AlphaFold structure, is of high interest. The function of the periplasmic domains in any CcsBA have not yet been fully determined, but some information exists in the literature. For example, there was limited analysis (4, 10 amino acid deletions) suggesting a potential role in cytochrome c synthase function for PD-1 (Sutherland et al. 2018. mBio: doi.org/10.1128/mbio.02134-18) and Cryo-EM identified two conformations that indicate a structural change in PD1 and PD2 upon heme interaction with the WWD domain (Mendez et al. 2022. Nat Chem Biol: doi: 10.1038/s41589-021-00935-y). To fully elucidate the role of PD-1 and PD-2, a comparative, in-depth analysis of the domains from multiple CcsBAs is warranted and of high importance. However, we feel that this is outside the scope of this manuscript and is among the future directions we plan to pursue.

2) Since the AF3 server is available, adding predictions of CcsBA with heme binding and comparing them with *H. hepaticus* structure would be a valuable addition to Figure 6.

Response: We thank the reviewer for this excellent suggestion. We have performed the new analysis and incorporated the results into the manuscript. For consistency, AlphaFold 3 predicted structures are used throughout the manuscript. Reduced size versions of revised figures are provided here for ease of review:

- *Fig. 2d, e, f, g – AlphaFold 3 predicted structures without heme replaced AlphaFold 2 predictions*

- Supplementary Fig. 2a, c – pLLDT for AlphaFold 3 predictions

- Fig. 6b, c – AlphaFold 3 was used to predict protein-heme interactions and heme was modeled into the TM-heme domains of *H. pylori* and *C. jejuni*. Discussion of AF 3 and TM-heme was added. Please see pg 19, line 404:

“AlphaFold 3 can predict protein-molecule interactions⁴⁰ and was used to predict heme interaction with *H. pylori* and *C. jejuni* CcsBA. These structural predictions were compared with the *H. hepaticus* CcsBA cryo-EM and show positioning of TM-His1,2 is similar, supporting the chemical complementation data that these residues act as axial heme ligands (Fig. 6a-c).”

- Fig. 6d, e, f, g – AlphaFold 3 predicted structures replaced AlphaFold 2 predictions

- Supplementary Fig. 10b, c – AlphaFold 3 was used to predict protein-heme interactions in the WWD domain of *H. pylori* (S10b) and *C. jejuni* (S10c). These were compared to the *H. hepaticus* WWD-heme interaction determined by cryo-EM (S10a). Note that the *H. pylori* WWD-heme interaction prediction did not correspond well with our biochemical cysteine/heme crosslinking results nor the *H. hepaticus* cryo-EM determined positioning, thus AF 3 WWD-heme predictions were included as a Supplementary Figure. A discussion of heme modeling into the WWD domain was added to the manuscript. Please see pg 21, line 434:

“AlphaFold 3⁴⁰ predicted heme interaction with *H. pylori* and *C. jejuni* CcsBA WWD domains (Supplementary Fig. 10b, c) were compared to heme positioning in *H. hepaticus* CcsBA determined by cryo-EM (Supplementary Fig. 10). *C. jejuni* predicted heme positioning is similar to *H. hepaticus* with both heme vinyl groups located near a cysteine/heme crosslinking residue. *H. pylori* predicted heme positioning did not correspond to the *H. hepaticus* CcsBA heme positioning (Supplementary Fig. 10b), thus confirmation of heme vinyl group positioning in the WWD domains of *H. pylori* and *C. jejuni* CcsBA awaits structural determination. The cysteine/heme crosslinking results can also provide insight.....”

- Material and Methods have been updated. Please see pg 23, line 493:

“CcsBA structural modeling. The *H. pylori* (strain ATCC 700392/26695) CcsBA and *C. jejuni* (strain ATCC 700819/NCTC 11168) CcsBA predicted structures were obtained using the AlphaFold 3⁴⁰ launched on 2024-05 to predict protein structures and protein-ligand (heme) interactions. For both the organisms, most of the predicted structures indicated a high level of model confidence (per residue confidence score, pLDDT, see Supplementary Fig. 2a, c). The PDB files of *H. pylori* & *C. jejuni* (obtained from AlphaFold 3 prediction) and *H. hepaticus* CcsBA open (PDB: 7S9Y¹⁶) conformation were uploaded to PyMOL (version 2.5.2). Then these structures were superimposed to give the root mean standard deviation (RMSD) values between them.”

3) While it is enticing to suggest that the WWD domain could serve as a novel therapeutic target, the lack of supporting data necessitates caution. The author should consider either removing this section from the conclusion or providing preliminary data. For example, demonstrating how a compound might bind to the WWD motif and inhibit cytochrome c maturation could strengthen the statement.

Response: We thank the reviewer for this comment. Respectfully, we feel that a discussion of the WWD as a novel therapeutic target is important, especially in context of the conservation of WWD-heme interaction determined here, which could potentially improve the design of new compounds. We have added a discussion of inhibition of System II and I by metal porphyrins to establish that inhibition of these pathways is feasible – we thank the reviewer for this comment, as it reminded us of this accidental oversight. However, as we do not have preliminary data for inhibition via the WWD domain specifically, we have revised this section to make clear that it is a proposed therapeutic target. Please see pg 22, line 457:

“The WWD domain as a potential novel therapeutic target

*The conservation of heme binding residues in the cytoplasmic TM-heme site and the periplasmic WWD/P-His heme domain determined here are important to our basic biological understanding of cytochrome c synthase function. Conservation of heme binding in the active site of CcsBA from multiple bacteria leads to the proposal that CcsBA could be a target for novel inhibitors. Early studies showed that addition of exogenous alternative metal porphyrins (e.g., ZnPPiX) inhibited cytochrome c biogenesis by recombinantly expressed System II and System I pathways⁵⁴, demonstrating that inhibition of these pathways is feasible. CcsBA is an attractive target as it is essential in pathogens including *H. pylori* and *C. jejuni*^{18,19,24,25,28–31}, thus targeting the synthase could inhibit pathogen growth. Additionally, differences in substrate recognition exist between CcsBA and the human cytochrome c synthase, HCCS, therefore cross-reactivity is unlikely. For example,....”*

Minor points:

1) L99, structural predictions are not “determined”, should only be used for experimental determination.

Response: We thank the reviewer for helping to improve our terminology. We have changed “determined” to “generated” to clarify this point for the reader. Please see pg 7, line 124.

2) L189, missing percentage number?

Response: We apologize for the confusion and have clarified this section. Please see pg 11, line 205: "...co-purify with ~wild type..." was changed to "...co-purify with approximately wild type..."

3) L242, The Ilcu et al 2023 CcmABCD/E paper should be cited.

Response: Thank you for catching this oversight. The reference has been added. Please see pg 13, line 261.

4) L285, Figure 4b and supplementary Figure 7d are identical. Same problem in Figure 5b and supplementary Figure 8d

Response: The reviewer is correct, Fig 4b is identical to original Supplementary Fig 7d, membrane 1 and Fig 5b is identical to original Supplementary Fig 8d, membrane 1. The purpose of the supplementary figures is to show that a) two membranes were layered to capture any 'free' heme that passed through membrane 1 during the transfer step and b) free heme from both membranes was quantified and used to determine the bound to free heme ratios. Thank you for identifying this point of confusion. To clarify this point for the reader, the supplemental figure legends were modified. Please note, in response to Reviewer 3, pt 11 original Supplementary Fig 7d was expanded to include discussion of the E842C breakdown product, so the numbering has changed:

Supplementary Fig 7g, h) b-type (free) heme can transfer through a 0.2 μ M nitrocellulose membrane, therefore 2 membranes are layered for all heme stains and heme from both membranes is quantified to determine the final ratio. Note, Membrane 1 alone is presented in Fig 4b for simplicity and Membranes 1 and 2 are presented here to demonstrate free heme transfer.

Supplementary Fig 8d) As described in Supplementary Figure 7, heme from both membranes is quantified to determine the final ratio in these experiments. Note, Membrane 1 alone is presented in Fig 5b for simplicity and Membranes 1 and 2 are presented here to demonstrate free heme transfer.

5) Supplementary Figure 9, quantify the data and present it also in histogram similar to that in Figure 3.

Response: The triplicate data displayed in Supplementary Figure 9 was quantified and presented in Figure 5h in the main text. To clarify this point for the reader, the Supplementary Figure 9 legend was modified:

"Supplementary Figure 9. Synthase function of C. jejuni CcsBA WWD cysteine variants. As described in Figure 5, the WWD variants were co-expressed with cytochrome c_4 in E. coli Δ ccm to determine synthase function. Representative triplicate experiment is shown. Each membrane/heme stain contains WT and vector controls for normalization during quantification. Heme stains were quantified to determine synthase function with WT normalized to 100%.

Quantification is displayed in Figure 5h. Single samples of WT, vector and each cysteine variant were selected to generate Figure 5g.”

Reviewer #3 (Remarks to the Author):

The manuscript entitled “Helicobacter pylori and Campylobacter jejuni bacterial holo-cytochrome c synthase...” by Yeasmin, T. et al., is on the heme binding sites of the CcsBA protein(s) from these species with System II cytochrome c biogenesis. The authors provide experimental support for the functions of structurally conserved features of these proteins using biochemical approaches. The experimental work was carried out very well, and the conclusions are well supported. However, similar studies (Ref 15) and the cryoEM structure (ref 16) of the CcsBA complex from Helicobacter hepaticus, a homologous species, have been reported earlier. Although this study could be seen as a repetition of what has been done earlier with H. hepaticus, it might be argued that providing experimental support for functional conservation of predicted structures is important in the case of H. pylori and C. jejuni as this information may eventually provide more reliable therapeutic target(s), making this study worthy of publication.

The following mostly minor modifications may further improve the manuscript.

1- Fig. 1 legend: for clarity, mention that CcsBA is a single protein that undergoes “natural proteolysis” yielding the CcsB and CcsA components as shown and add a reference.

Response: Thank you for this suggestion. We have modified the legend as follows. Please see pg 3, line 44:

“Figure 1. Schematic of System II. System II, CcsBA (blue), is a bifunctional enzyme, which is proposed to transport heme (red) across the bacterial membrane and attach it to cytochrome c (yellow) at a conserved CXXCH motif. CcsBA is encoded as a single ORF that undergoes “natural proteolysis” resulting in CcsB and CcsA components⁵ as shown.”

2- p. 4, l.57: To help the reader, perhaps add a sentence describing the basis of the computational approach used to evaluate the structural conservation between the predicted structures (e.g., superimposition of a cryoEM derived structure with those computationally (Rosetta, AlphaFold...) predicted structures, and determination of the RMSD values around the conserved features.

Response: Thank you for this suggestion, we agree that more detail would assist the reader with understanding the structural comparisons and methods used. We have modified the manuscript as follows:

- See pg 4, line 58 “More recently, a computational study used RoseTTAFold²⁷ to evaluate the predicted structures ...”

- See pg 7, line 138: “Superimposition of the *H. hepaticus* cryo-EM structure with the computationally predicated *H. pylori* CcsBA AlphaFold 3 structure indicates a high level of structural conservation (RMSD 1.842 for overlay of complete structures), including positioning of the 14 TM domains and the PD1 and PD2 domains (Fig. 2f). Superimposition of the computationally predicated *C. jejuni* CcsBA AlphaFold 3 structure with the *H. hepaticus* CcsBA cryo-EM structure indicates structural conservation in positioning of the 14 transmembrane domains (RMSD 1.202 for overlay of complete structures),....”
- See pg 7, line 119. Figure 2 legend was modified to also include RMSD: “f) Superimposition of *H. hepaticus* cryo-EM structure (deep purple) and *H. pylori* predicted structure (deep teal), RMSD 1.842. g) Superimposition of *H. hepaticus* cryo-EM structure (deep purple) and *C. jejuni* predicted structure (forest green), RMSD 1.202.”
- See pg 23, line 497. “...The PDB files of *H. pylori* & *C. jejuni* (obtained from AlphaFold 3 prediction) and *H. hepaticus* CcsBA open (PDB: 7S9Y¹⁶) conformation were uploaded to PyMOL (version 2.5.2). Then these structures were superimposed to give the root mean standard deviation (RMSD) values between them.”

3- p. 4, l.69: clarify “...CcsB CcsA have not been functional...”; I guess the authors mean that: the recombinant expression of the CcsB and CcsA as two separate proteins in *E. coli* has not yielded a functional CcsBA complex (if known, due to some reason like not assembled, not stoichiometric, not stable, not soluble, etc etc..), unlike CcsBA as a single fused ORF.

Response: Thank you. To clarify this point, the section has been revised. Please see pg 4, line 69:

“...The single-fused ORF CcsBA has been amenable to recombinant studies and is stable and functional in *E. coli*^{5,32}. In contrast, recombinant expression of System II encoded as two separate proteins (CcsB CcsA) in *E. coli* has not resulted in a functional CcsBA complex (e.g.³³), limiting the ability to perform structure-function analyses...”

4- p. 5, l.71: “... lack of analysis ... a major gap”: perhaps add “...lack of experimental data...”

Response: Thank you for this suggestion, we have added it to the manuscript. Please see pg 4, line 72:

“...*H. hepaticus* CcsBA has proven to be an excellent model for characterization of CcsBA, however the lack of experimental data for CcsBAs from other organisms has been a major gap in the field.”

5- Figure 2 and other figures: a- please make consistent the figure legends and related text using capital letters with the figure panels using small cases. b- if possible, provide an overall (or regional) RMSD value for deviations between the compared structures to give the reader a feeling about how good is the conservation between the structures. For easier visualization, is it possible to color the heme moieties on *H. hepaticus* in black, or point them out by arrows? Also, indicate the β -cap region of PD1 with thicker dotted line to ease noting its location on the structures?

Response:

a – We have corrected this throughout the text and thank the reviewer for catching this oversight.

b – Thank you for these suggestions to improve the clarity of our figures. Included below is a reduced size version of revised Figure 2 for ease of review. The following improvements were made:

- *H. hepaticus* hemes are indicated with arrows and labeled. Figure legend has been updated to include the following. Please see pg 7, line 113:

“Heme in red and indicated by arrow with heme interaction domains labeled”.

- β -cap region of PD1 line has been thickened and changed to a dark gray color

- As described in the response to pt #2, RMSD values were calculated and included in the main text and Figure 2 legend; AlphaFold pLDTT figures were added to Supplementary Fig 2.

6- Just a curiosity: can the extra ~100 AAs long beta-sheet portion of *C. jejuni* PD1 domain be deleted (at least in *E. coli*) and if so, is it functional? Is it plausible that CcsBA is interacting with another protein in *C. jejuni*?

Response: We agree that the extended beta-sheet domain predicted in *C. jejuni* is of high interest. As described in our response to Reviewer 2, Major pt 1 we feel an in-depth structure-function analysis of the PD1 and PD2 domains is warranted, however outside the scope of this paper and among the future directions our group is pursuing. Regarding CcsBA interacting with another protein in *C. jejuni* – that is a possibility, but if an interaction does occur we predict it is not essential for CcsBA function. *C. jejuni* CcsBA was functional in recombinant *E. coli*, therefore it is unlikely that a *C. jejuni* specific protein association with the PD1 extension is required for heme attachment.

7- p. 9, l.162: to support this suggestion, has any mutagenesis of the predicted proteolytic cleavage site been performed in any species, and what was the outcome?

Response: Thank you for this suggestion. We have added additional information regarding natural proteolysis of CcsBA from the literature. Please see pg 9, line 174:

*“...unlikely to be an experimental artifact. In fact, a biological role for proteolysis is supported as multiple strategies to eliminate natural proteolysis were unsuccessful. For example, neither expression of *H. hepaticus* CcsBA in a panel of protease deficient *E. coli* strains nor a 20 amino acid deletion encompassing the site of natural proteolysis prevented the observed proteolysis¹⁵.”*

8- Fig. S3, legend, Table S1: clarify the genotype(s) of MS1175 and MS1025 (wt or mutant)

Response: We have clarified the Supplementary Figure 3 legend to include a description of plasmids in MS1175 and MS1025:

*“.....To ensure the low synthase activity of *H. pylori* was not due to a mutation in the recombinant *E. coli* strain, two independently constructed strains were used (MS1175, MS1025 – *E. coli* Δ ccm with cytochrome *c*₄ (pRGK334) and wild type *H. pylori* GST:CcsBA (pMCS1075)) and assay was run in duplicate (lanes 7/8, 9/10).”*

9- p. 10, l.205 on: a- Considering that *H. pylori* (I assume wt) CcsBA activity cannot be determined using *B. pertussis* cytochrome *c*₄, then what is the proof that the function of the 4 His conserved in this species? Is it by inference to *C. jejuni* and *H. hepaticus*? If so, clearly state this point. b- Could a cytochrome distinct from *c*₄ be more suitable as a substrate for *H. pylori*?

Response:

*-a, Yes, the reviewer is correct we should more clearly state that *H. pylori* conserved histidine function is based on the results from other CcsBAs. We have clarified this in the following sections:*

- See pg 12, line 223. To clarify *H. pylori* functional assays were performed with wildtype “...two independently constructed wildtype *H. pylori* CcsBA functional strains...”*
- See pg 19, line 397. Additional discussion of *H. pylori* TM-His in Conclusion “...Further, TM-His_{1,2} in *C. jejuni* CcsBA act as axial heme ligands (Fig. 3g, k). Experimental evidence that the *H. pylori* CcsBA TM-His_{1,2} are required for synthase function and act as axial heme ligands was not obtained due to low levels of synthase function (Supplementary Fig. 3h, i). However, based on conservation of TM-His_{1,2} and the experimental evidence from *H. hepaticus* CcsBA^{5,15}, *W. succinogenes* CcsBAs⁴³, *Chlamydomonas reinhardtii*^{21,42} and *C. jejuni* CcsBA (Fig. 3e, g) we predict *H. pylori* TM-His_{1,2} will function as axial ligands to heme.”*

*-b, Yes it is likely that a different cytochrome *c* would be a suitable substrate for *H. pylori*. Future directions include constructing a panel of cytochromes *c* to identify a substrate for *H.**

pylori CcsBA

10- p. 11, l.215: a- add a reference to “cytoplasmic heme vestibule”? b- l. 227, ‘imidazole’ spelling?

Response: Thank you for catching this, both points have been corrected –

-a, See pg 12, line 234: “... (alternatively called the cytoplasmic heme vestibule^{16,17}).”

-b, See pg 12, line 246: “...robustly chemically complemented with imidazole^{5,45}.”

11- Fig. 4 and Fig. S7: What is the heme-stained band visible with E842C mutant running between CcsA and GST*, and what is GST*?

Response: We have added the following statement to Figure legends 3, 4, 5, S7, S8 to clarify the labels for the reader: “CcsBA purifies as two major polypeptides: GST:CcsB’ and ‘CcsA.’ – indicates natural proteolysis. GST* - proteolyzed GST.”. Thank you for pointing this out.

E842C – the heme-stained band is proteolyzed protein containing the crosslinked heme in the WWD domain. This breakdown product can also be seen via faint heme staining bands W827C and W836C. We apologize for not stating that originally. Reduced size version of revised figures are included below for ease of review. To make this point clear to the reader we have done the following:

- Fig. 4a, b: added a *CcsA label to indicate the proteolyzed fragment
- See pg 15, line 319. Fig. 4 legend: added description of “*CcsA”: “*CcsA – proteolyzed polypeptide containing the WWD domain with crosslinked heme”
- See pg 15, line 305. Main text of manuscript: “*H. pylori* CcsBA affinity purified proteins have a degradation product that contains the WWD polypeptide as indicated by the heme-stained band (Fig. 4b *CcsA, Supplementary Fig. 7d-h). Note that covalently bound heme (i.e., from formation of a cysteine/heme crosslink) is retained at the protein polypeptide upon SDS-PAGE, therefore the heme-stained *CcsA band contains the WWD domain with crosslinked heme.”

- New Supplementary Fig. 7d-h. To clearly distinguish *CcsA and GST* protein polypeptides a supplemental figure containing a Coomassie, GST immunoblot, heme stain (membranes 1 and 2) and an inverted heme stain image to show faint breakdown product in W827C and W837C was added.
- New Supplementary Fig. 7d-h legend: **“Supplementary Figure 7. Cysteine/heme crosslinking in the *H. pylori* CcsBA WWD domain. Each residue of the *H. pylori* CcsBA WWD domain was individually mutated to cysteine. The single amino acid variants were affinity purified in groups, each with a WT control. Groups were analyzed by a) Coomassie total staining and b) anti-GST immunoblotting. CcsBA purifies as two major polypeptides: GST:CcsB’ and ‘CcsA . ‘ – indicates natural proteolysis. *CcsA – proteolyzed polypeptide containing CcsA WWD domain. GST* - proteolyzed GST. c) Heme stains were quantified by determining the ratio of CcsA bound heme to b-type (free) heme from membrane 1 and membrane 2. Variants were compared to a WT that was induced and purified at the same time. Variants with a ratio of >2 for CcsA bound to b-type heme were further assessed. d-h) To demonstrate the *CcsA heme-stained polypeptide is distinct from GST* the following were aligned by the MW markers. d) Coomassie to show total protein. Note, this Coomassie is also presented in Fig 4a. e) α-GST immunoblot. f, g) Heme stain. Note b-type (free) heme can transfer through a 0.2 μM nitrocellulose membrane, therefore 2 membranes are layered for all heme stains and heme from both membranes is quantified to determine the final ratio. Note, Membrane 1 alone is presented in Fig 4b for simplicity and Membranes 1 (f) and 2 (g) are presented here to demonstrate free heme transfer. h) Inverted image of the heme stain of Membrane 1 to demonstrate low-levels of heme stained bands in W827C and W836C.”**

12- p. 14, l.299-303: a- Clarify and modify the sentence: if the amount of crosslinked-heme in the WWD domain is very small, even this perturbs the TM-heme environment, UV-Vis spectra may not be sensitive enough to show it. b- is it possible to detect by pyridine/hemochrome the XL-heme derivatives, which should act as a cyt c? As this method would get rid of cyt b, could it provide a better means of quantification for CcsA~XLheme than that used here (ratio)?

Response: Thank you for these suggestions.

-a – We have modified the section. Please see pg 16, line 326:

“It is known that CcsBA retains most of the heme in the cytoplasmic TM-heme site¹⁶, thus this b-heme is predicted to account for the majority of the heme signal. Therefore, even if the small amount of crosslinked heme in the WWD domain results in perturbation of the TM-heme environment, UV-vis spectra may not be sensitive enough to detect these changes.”

-b We thank the reviewer for this suggestion, however pyridine assays were previously unable to detect the crosslinked heme in H. hepaticus CcsBA likely because the majority of heme that co-purifies with the protein is the TM located b-heme (as indicated by the 556 alpha-peak) (Sutherland et al. 2018. mBio: doi.org/10.1128/mbio.02134-18). Thus, these assays were not pursued in this study.

13- p. 17, l.346, please recheck carefully the number of the residues mentioned in the text and in Fig. 5 (W893 or W983? S978 or S988?).

Response: We have checked and fixed all residue numbers in the text. We apologize for the confusion.

14- Introduction: Could you better explain why Ccs system is essential for H. pylori and C. jejuni, and this under which specific growth conditions? Is this due to the lack of the cytochrome c containing respirator pathway with a single cbb3-type cytochrome oxidase and absence of any other only b-type cytochrome containing alternative respiratory branch? Is CcsBA also essential in H. hepaticus, and if not explain why? Perhaps emphasizing this point might add more value to this work and save it from appearing as a repetition of what has been done earlier?

Response: Thank you for this suggestion, it will provide additional context to reader. To our knowledge, deletion of H. hepaticus CcsBA has not been attempted. We added the following to the introduction. Please see pg 5, line 85:

“H. pylori only encodes a cytochrome c oxidase for growth in the presence of oxygen and a ccsBA deletion could not be obtained²⁸. While C. jejuni encodes a cytochrome c oxidase and cytochrome bd for growth in the presence of oxygen²⁸, yet ccsBA deletions were still not obtained under microaerobic growth conditions³¹. Thus, demonstrating that ccsBA is an essential gene in both these organisms^{28,31}. Therefore, recombinant expression is required for genetic studies”

15- Fig. 5c: Explain why two bands correspond to cyt c₄ of *B. pertussis* (12 and 24 kDa, dimerization or proteolysis?) seen in this and other figures.

Response: We apologize for not clearly explaining the results of this assay. The initial development of this assay demonstrated endogenous proteolysis of cytochrome c₄ to a 12 kDa fragment by heme stain and the proteolysis was confirmed by mass spec (Feissner et al. 2006: doi:10.1111/j.1365-2958.2006.05132.x). We have added the following to the main text and Fig. 3 to clarify for the reader.:

- *Main text revision, please see pg 11, line 218:*

*“...recombinantly co-expressed with *Bordetella pertussis* cytochrome c₄ in an *E. coli* strain lacking the endogenous cytochrome c biogenesis genes (*E. coli* Δccm) and synthase function (i.e. heme attachment to cytochrome c) was monitored via an enhanced chemiluminescence (ECL)-based heme stain⁴⁵. Note, this assay results in full-length holocytochrome c₄ (24 kDa) and an endogenously proteolyzed 12 kDa fragment³².”*

- *Fig 3 legend revision, please see pg 10, line 186:*

“... via SDS-PAGE and heme stain of holocytochrome c (24 kDa – full length; 12 kDa – endogenously proteolyzed). Quantification...”

No obvious issue with statistics and reproducibility of the work.

REVIEWERS' COMMENTS:

Reviewer #1 (Remarks to the Author):

All review comments have been reacted to adequately.

Reviewer #2 (Remarks to the Author):

The authors have addressed all my concerns and have added additional analysis. The paper is ready to be published in its current form.

Reviewer #3 (Remarks to the Author):

This is a re-review of a revised version of this manuscript; the initial manuscript was deemed worth of publication at that time, and the minor points raised to further improve the manuscript are now addressed fully and satisfactorily by the authors. Therefore I have no further objection to the acceptance and publication of this worthwhile piece of study.